# Functional Selectivity of Cannabinoid Type 1 G Protein-Coupled Receptor Agonists in Transactivating Glycosylated Receptors on Cancer Cells to Induce Epithelial–Mesenchymal Transition Metastatic Phenotype

**DOI:** 10.3390/cells13060480

**Published:** 2024-03-08

**Authors:** David A. Bunsick, Jenna Matsukubo, Rashelle Aldbai, Leili Baghaie, Myron R. Szewczuk

**Affiliations:** 1Department of Biomedical and Molecular Sciences, Queen’s University, Kingston, ON K7L 3N6, Canada; 17dab5@queensu.ca (D.A.B.); 18jem9@queensu.ca (J.M.); 19ra57@queensu.ca (R.A.); 16lbn1@queensu.ca (L.B.); 2Faculty of Medicine, University of Ottawa, Roger Guindon Hall, 451 Smyth Rd #2044, Ottawa, ON K1H 8M5, Canada

**Keywords:** cannabinoid, CB1 receptor, Neu1, MMP9, SEAP, RAW-blue, PANC-1, SW-620

## Abstract

Understanding the role of biased G protein-coupled receptor (GPCR) agonism in receptor signaling may provide novel insights into the opposing effects mediated by cannabinoids, particularly in cancer and cancer metastasis. GPCRs can have more than one active state, a phenomenon called either ‘biased agonism’, ‘functional selectivity’, or ‘ligand-directed signaling’. However, there are increasing arrays of cannabinoid allosteric ligands with different degrees of modulation, called ‘biased modulation’, that can vary dramatically in a probe- and pathway-specific manner, not from simple differences in orthosteric ligand efficacy or stimulus-response coupling. Here, emerging evidence proposes the involvement of CB1 GPCRs in a novel biased GPCR signaling paradigm involving the crosstalk between neuraminidase-1 (Neu-1) and matrix metalloproteinase-9 (MMP-9) in the activation of glycosylated receptors through the modification of the receptor glycosylation state. The study findings highlighted the role of CB1 agonists AM-404, Aravnil, and Olvanil in significantly inducing Neu-1 sialidase activity in a dose-dependent fashion in RAW-Blue, PANC-1, and SW-620 cells. This approach was further substantiated by findings that the neuromedin B receptor inhibitor, BIM-23127, MMP-9 inhibitor, MMP9i, and Neu-1 inhibitor, oseltamivir phosphate, could specifically block CB1 agonist-induced Neu-1 sialidase activity. Additionally, we found that CB1 receptors exist in a multimeric receptor complex with Neu-1 in naïve, unstimulated RAW-Blue, PANC-1, and SW-620 cells. This complex implies a molecular link that regulates the interaction and signaling mechanism among these molecules present on the cell surface. Moreover, the study results demonstrate that CB1 agonists induce NFκB-dependent secretory alkaline phosphatase (SEAP) activity in influencing the expression of epithelial–mesenchymal markers, E-cadherin, and vimentin in SW-620 cells, albeit the impact on E-cadherin expression is less pronounced compared to vimentin. In essence, this innovative research begins to elucidate an entirely new molecular mechanism involving a GPCR signaling paradigm in which cannabinoids, as epigenetic stimuli, may traverse to influence gene expression and contribute to cancer and cancer metastasis.

## 1. Introduction

The gaps in our understanding of how CB cannabinoids are implicated in host metabolism via their G protein-coupled receptors (GPCRs) emphasize the need for research to elucidate the mechanistic action of these molecules in the body. The details and specific mechanisms of the effects of CB consumption by individuals on energy homeostasis and host metabolism remain to be determined. This topic is highly relevant as CB is an option to improve health and pain. However, its consumption has been associated with increased risk factors for metabolic disorders and drug side effects.

G protein-coupled receptors are seven-transmembrane-domain proteins that mediate physiological responses to hormones, neurotransmitters, and environmental stimulants [1]. As a result, GPCRs are the most important targets for drug discovery, with ~40% of all FDA-approved drugs acting on at least one member of the GPCR gene family [2]. Interestingly, in the mid-1990s, different GPCR agonists were discovered to be ‘biased’ towards a specific downstream signaling pathway [3]. This concept, called biased agonism or functional selectivity, has created new opportunities for developing drugs with more useful therapeutic profiles.

GPCRs comprise the most prominent family of cell surface signal-transduction molecules in mammalian cells. They have long been implicated in the transactivation of receptor tyrosine kinases (RTKs) in the absence of a respective growth factor, particularly for receptors that bind the epidermal growth factor, platelet-derived growth factor, fibroblast growth factor, and neurotrophins [4]. Delcourt and colleagues [5] provided an eloquent review of the mechanisms involved in this novel cross-communication between GPCRs and RTKs. Reciprocally, growth factors binding to RTKs can also utilize GPCR-signaling molecules to initiate the molecular organizational signaling platform of novel Neu1 and MMP-9 crosstalk in alliance with RTK on the cell surface. This novel GPCR-signaling platform was identified to be critically essential for neurotrophin factor-induced TrkA [6], insulin [7,8,9,10], epidermal growth factor (EGF) [11], and Toll-like (TLR) [12,13,14] receptor activation and cellular signaling. These findings have revealed a novel concept in which GPCR activation is essential for growth-factor RTK and pathogen-sensing TLR activation. The mechanism(s) here involves the formation of a functional signaling complex between the GPCR-RTK and GPCR-TLR partners. However, GPCR transactivation by RTK ligands or by TLR pathogen-associated molecular patterns has been established for a few GPCR-RTK or GPCR-TLR partners, as reviewed previously [12]. Haxho et al. [10] have reported on the structure of the insulin receptor, outlining the importance of insulin receptor glycosylation and its modifications and the critical molecules in the insulin Irβ receptor’s activation and the subsequent cell signaling. Unexpectedly, other GPCR agonists, such as bradykinin (BR2) and angiotensin 2 receptor type I (AT1R), were also found to exist in a heteromeric GPCR complex, with GPCR neuromedin B (NMBR), Irβ, and Neu1 in naïve and stimulated cells overly expressing the insulin receptor [8].

Interestingly, Rozenfeld et al. [15] reported a similar AT1R–cannabinoid CB1R heterodimerization complex, revealing a new mechanistic action for the pathogenic properties of angiotensin-2. Also, the cannabinoid Δ9-tetrahydrocannabinol (THC) was found to disrupt human epidermal growth factor receptor 2 (HER2)–CB2R complexes by selectively binding to CB2R, leading to the inactivation of HER2 through the disruption of HER2–HER2 homodimers, and the subsequent degradation of HER2 by the proteasome via the E3 ligase c-CBL [16]. Wager-Miller et al. [17] reported that the ability of the CB1 receptors to form heterodimers increases their potential to generate biased downstream signaling effects. For example, CB1 receptors can also heterodimerize with various GPCRs, including the D_2_ dopamine receptor [18,19,20], µ-opioid receptor [21], A_2A_ adenosine receptor [22], and β_2_ adrenergic receptors (β_2_Ars) [23]. Of these receptor pairs, significant research has investigated the CB1-D_2_ heterodimer and its unusual mechanistic action. For example, several studies have suggested that the CB1-D_2_ heterodimer stabilizes a CB1 active state with increased coupling to Gαs over Gαi, leading to an accumulation of cAMP [18,19,20,24]. The Gs alpha subunit (Gαs) is a subunit of the heterotrimeric G protein Gs that stimulates the cAMP-dependent pathway by activating adenylyl cyclase [25]. Notably, the D2 receptor may only induce this switch toward Gαs coupling when it is in its active state [26]. In contrast to the effects of CB1-D_2_ heterodimerization, the physical interaction of CB1 with β_2_AR is proposed to bias CB1 signaling away from Gαs and toward Gαi, leading to an increase in ERK1/2 phosphorylation [23,26]. Overall, the concept of heterodimerization significantly adds to the complexity of CB1 functional selectivity as it depends not only on the identities of the GPCRs present in the complex but also on the active states of each of the receptors [26]. Further expanding this CB1 functional selectivity complexity, the CB1 GPCR receptors can also form higher-order oligomers, such as the CB1/D_2_/A_2A_ hetero-oligomers [27]. As existing research demonstrates the ability of dimerization to induce biased signaling effects, oligomerization is likely to increase the topic diversity of functional selectivity exponentially with the CB1 cannabinoid receptor.

Amidst the interest in cannabis usage, there is an increase in research on the impacts of endocannabinoids (Ecs), cannabinoids, and associated CB1/CB2 receptors on tumor development and metastasis [28]. Of the existing literature, evidence has revealed that cannabinoids can impact markers of epithelial–mesenchymal transition (EMT), one of the hallmarks of cancer metastasis [29,30,31,32]. However, there exists substantial variability in the responses of cancer cells to cannabinoids in the context of invasiveness and metastatic capacities.

To this end, the present study investigated the involvement of CB1 GPCRs in a novel biased GPCR signaling paradigm involving the crosstalk between neuraminidase-1 (Neu-1) and matrix metalloproteinase-9 (MMP-9) in the activation of the glycosylated RTK and TLR receptors through the modification of the receptor glycosylation state for downstream signaling. The study results also reveal that CB1 agonists can induce NFκB-dependent secretory alkaline phosphatase (SEAP) activity in inducing the expression of epithelial–mesenchymal markers, E-cadherin, and vimentin in SW-620 cells.

## 2. Materials and Methods

### 2.1. Cell Lines

Three cell lines were used in this study: PANC-1 (ATCC^®^ CRL-1469™), RAW-Blue macrophages (InvivoGen, San Diego, CA, USA), and SW-620 (ATCC^®^ CCL-227™) cells. RAW-Blue™ cells (Mouse Macrophage Reporter Cell Line, InvivoGen) derived from RAW 264.7 macrophages were grown in a culture medium containing Zeocin as the selectable marker [33]. They stably express a secreted embryonic alkaline phosphatase (SEAP) gene inducible by NF-κB and AP-1 transcription factors. Upon stimulation, RAW-Blue™ cells activate NF-κB and AP-1, leading to SEAP secretion, detectable and measurable using QUANTI-Blue™ (InvivoGen) SEAP in the medium. RAW-Blue™ cells are made to be resistant to Zeocin™ and G418 antibiotics. The cells were grown in conditioned media 1 × Dulbecco’s modified eagle medium (DMEM, Gibco, Rockville, MD, USA), with 10% fetal bovine serum (FBS) (HyClone, Logan, UT, USA) and 5 μg/mL plasmocin (InvivoGen, San Diego, CA, USA). They were maintained at 5% CO_2_ and 37 °C.

### 2.2. Reagents

The CB1 agonists, N-arachidonoyl aminophenol (AM 404) (KI: 2.57 μg/mL) [34], Arvanil (KI: 1.28 μg/mL) [35], and Olvanil (KI: 4.127 μg/mL) [35], were purchased from Alomone Labs, Jerusalem BioPark (JBP) (Jerusalem, Israel) and utilized in a dose-dependent manner. The KI value was used to create the dose–response curve. The saturation value is a 3:1 concentration of the KI value for each agonist. The subsequent doses are 1:10 and 1:30 dilutions of the KI value. In the live cell sialidase assay experiments, 2-(4-methylumbelliferyl)-α-D-N-acetylneuraminic acid (98% pure 4-MUNANA; Biosynth International Inc., Itasca, IL, USA), a sialidase substrate, was used at a concentration of 0.318 mM diluted in tris-buffered saline (TBS). NMBR inhibitor, BIM-23127, was used at 12.5 μg/mL and purchased from Tocris Bioscience, IO Centre Moorend Farm Avenue, Bristol, BS11 0QL, UK. Oseltamivir phosphate (OP) (>99% pure OP, batch No. MBAS20014A, Solara Active Pharma Sciences Ltd., New Mangalore-575011, Karnataka, India), a broad-range neuraminidase sialidase inhibitor using predetermined effective dosages, was used at 300 μg/mL. MMP-9 inhibitor I (MMP9-i), a potent reversibly selective inhibitor of MMP-9, was used at 12.5 μg/mL and purchased from GLPBIO Technology, 10292 Central Ave #205, Montclair, CA 91763, USA. TLR4 ligand lipopolysaccharide (LPS) was used at 5 μg/mL from *Serratia marcescens* and purified by phenol extraction, as per Sigma Aldrich, (MilliporeSigma Canada Ltd., Oakville, ON, Canada).

### 2.3. CB1 Agonist Treatment Protocol (Time and Dosage)

The agonists AM-404, Arvanil, and Olvanil were chosen based on their interactions with the CB1 receptor and binding affinity. AM-404 is an indirect agonist of the CB1 receptor by binding to the fatty acid amide hydrolase (FAAH) enzyme, preventing anandamide breakdown [36,37]. AM-404 has a K_i_ value of ~6.5 µM (2.57 µg/mL) [38,39,40,41]. Aravnil is a potent direct agonist of the CB1 receptor and prevents anandamide breakdown. The agonist has a K_i_ value of ~2.9 µM (1.27 µg/mL) [36,41,42,43]. Olvanil is also a direct agonist of the CB1 receptor and prevents anandamide breakdown. Olvanil is a weaker agonist with a K_i_ value of ~10.1 µM (4.21 µg/mL) [41,44,45]. These values were used to create a dose–response curve where the K_i_ value is a 1:3 dilution of the saturation dose. A 1:10 and 1:30 dilution was made from the saturation dose. The cells were treated with the cannabinoid for 24 h, as previously reported by others [46,47,48].

### 2.4. Antibodies

In the co-localization protocol, anti-mouse CB1 conjugated with Alexa Fluor 594 was purchased at R&D Systems (Minneapolis, MN, USA). It was used at a concentration of 1 μg/10^6^ cells to stain CB1 receptors. The anti-mouse Neu1 conjugated with Alexa Fluor 488 antibody (Santa Cruz Biotechnology, Dallas, TX, USA) was used at a concentration of 1 μg/10^6^ cells. For immunofluorescence staining, primary mouse monoclonal IgG antibodies for E-cadherin and vimentin were purchased from Santa Cruz Technologies and used at a 1:10 dilution from 200 μg stock. The secondary goat anti-mouse Alexa Fluor 488 antibodies (Santa Cruz) were used at a concentration of 1:1000 for the immunofluorescence predetermined standardized protocol.

### 2.5. Sialidase Assay

PANC-1, RAW-Blue macrophage, and SW-620 cells were cultured and individually grown on a 12 mm circular glass slide in a sterile 24-well tissue culture plate in a conditioned medium for 24 h [6,49]. Once cells reached approximately 70% confluence, they were serum-starved for 24 h. Subsequently, media were removed from the wells and cells were treated with 4-MUNANA substrate, followed by the treatment of CB1 agonists alone or the CB1 agonist in combination with an inhibitor at a predetermined concentration. Once Neu1 hydrolyzes the 4-MUNANA sialidase substrate, free 4-methylumbelliferone (4-MU) is formed and fluoresces at 450 nm (blue color) when excited at 365 nm. Fluorescent images were captured using epi-fluorescent microscopy (Zeiss Imager M2, 20× objective) after 3 min. The sialidase activity was represented by blue fluorescence surrounding the cells’ periphery. The mean fluorescence intensity of 50 different points surrounding the cell was quantified using Image J software, 1.5g, Java 1.8.0_345 (64-bit).

### 2.6. NF-kB Dependent Secreted Embryonic Alkaline Phosphatase (SEAP) Assay

Briefly, a cell suspension of 1 × 10^6^ cells/mL in the fresh growth medium was prepared, and 100 μL of RawBlue suspension of cells (~100,000 cells) was added to each well of a Falcon flat-bottom 96-well plate (Becton Dickinson, Mississauga, Ontario, Canada) [33]. Following varying incubation times, AM-404, Olvanil, and Arvanil CB1 agonists were added to each well in a dose-dependent manner either alone or in combination with the specific MMP9 inhibitor (MMP9i); oseltamivir phosphate (OP) and BIM-23127 (BIM23) were added to each well 1 h before stimulation with ligands. The plates were incubated at 37 °C in a 5% CO_2_ for 18–24 h, followed with QUANTI-Blue™ (InvivoGen) reagent solution as per the manufacturer’s instructions. Briefly, 160 μL of resuspended QUANTI-Blue solution was added to each well of a 96-well flat-bottom plate, adding 40 μL supernatant from the treated RAW-blue cells. Following the plate incubation for 60 min at 37 °C, the SEAP levels were measured using a spectrophotometer (Spectra Max 250, Molecular Devices, Sunnyvale, CA, USA) at 620–655 nm. Each experiment was performed in triplicate.

### 2.7. Co-Localization

PANC-1, RAW-Blue macrophages, and SW-620 cells were cultured and individually grown on 12 mm circular glass coverslips within a sterile 24-well plate in the conditioned medium for 24 h. Once cells reached approximately 70% confluence, they were serum-starved for 24 h. Cells were fixed with 4 μg/mL paraformaldehyde (PFA) for 24 h at a 5 °C refrigerator. Subsequently, cells were permeabilized with Triton-X for 5 min and then blocked with 4% BSA in 0.1% Tween-TBS in the fridge for 24 h. Cells were treated with conjugated antibodies for CB1 receptors and Neu1 in a cold room on a rocker for 48 h. To account for non-specific background fluorescence, cells treated only with AlexaFluor-488 or AlexaFluor-594-conjugated secondary antibodies were included as controls. Coverslips with stained cells were mounted on microscope slides onto 3 μL of DAPI fluorescent mounting medium. Slide images were observed using Zeiss M2 epi-fluorescent microscopy (40× magnification, Carl Zeiss Canada Ltd., M3B 2S6 Toronto, Canada), capturing images under green (488 nm) and red (594 nm) channels. The Pearson correlation coefficient quantifies protein co-localization in the acquired images, and the results were expressed as a percentage determined using AxioVision software, Rel. version 4.6. Pearson’s correlation coefficient measures the linear proximity and association between two variables. A correlation value of 0.5 to 0.7 between the two variables would indicate that a significant and positive relationship exists between the two.

### 2.8. Immunofluorescence Staining

SW-620 cells were cultured and individually grown in a sterile 24-well plate on 12 mm circular glass coverslips containing the conditioned medium for 24 h. Upon reaching approximately 70% confluence, cells were media-starved and incubated with predetermined indicated concentrations of CB1 agonists (AM 404, Olvanil, and Arvanil) in the designated wells for 24 h. Control wells were incubated with media without FBS. Cells were then fixed at 4% PFA (300 µL) and incubated in the cold for 24 h. To facilitate the binding of antibodies, cells were permeabilized with 0.2% Triton x100 (300 µL) for 5 min and then blocked with 4% BSA/Tween20/TBS (300 µL) for 24 h in the cold to prevent non-specific binding. Primary antibodies were added at a 1:10 dilution in 4% BSA/Tween20/TBS solution and incubated in the fridge for 24 h. Subsequently, the secondary antibodies were added at a 1:1000 dilution in PBS and incubated in the fridge for 24 h. Cells treated with secondary antibodies were employed as a negative control. Stained cells on coverslips were mounted with 3 μL DAPI mounting media. Slides were visualized using epi-fluorescent microscopy (40× magnification), and illustrations were captured under a 488 nm green channel. The fluorescence of each marker was calculated using Corel Photo-Paint, where an average of eight points on the image was averaged and subtracted by the background fluorescence. The difference was multiplied by the pixel density.

### 2.9. Statistics

For statistical analysis, we used GraphPad Prism 10. Comparisons between groups from two independent experiments were conducted using a one-way analysis of variance (ANOVA) at 95% confidence, followed by Fisher’s uncorrected LSD multiple comparisons post hoc test with 95% confidence. Asterisks denote statistical significance.

## 3. Results

### 3.1. CB1 G Protein-Coupled Receptor Agonists Dose-Dependently Induce Neu1 Sialidase Activity in RAW-Blue Macrophage Cells

We recently reported a receptor signaling paradigm involving receptor ligand-induced GPCR-signaling via Gαi-proteins, MMP-9 activation, and the induction of Neu1 activation [13]. Central to this process was that the Neu1–MMP-9 complex is tethered at the ectodomain of TLR-4 receptors on the cell surface of naïve primary macrophages and TLR-expressing cell lines. Furthermore, we reported that a similar signaling paradigm was identified to be critically essential for neurotrophin factor-induced TrkA [6], insulin [7,8,9,10], EGF [11], and TLR [12,13,14] receptor activation and cellular signaling.

Collectively, ligand binding to its receptor may induce allosteric conformational changes in the receptor, which, in turn, potentiates GPCR-signaling and MMP-9 activation to induce Neu1 sialidase, as depicted in Figure 1A. The question is whether CB1 GPCR agonists binding to their respective GPCRs would directly induce Neu1 sialidase activity in a macrophage cell line in the absence of TLR-specific ligands. The data in Figure 1B revealed that CB1 GPCR agonists AM-404, Arvanil, and Olvanil at the indicated concentrations on live RAW-Blue cells significantly and dose-dependently induce sialidase activity. There were insignificant differences in sialidase activity when the agonists exceeded their inhibitor constant KI value, signifying that the sialidase activity induced by the CB1 agonists has reached saturation. We also found that the average fluorescence decreases significantly followed a 1:10 dilution of the agonist. Notably, the fluorescent emission of 4-MU is produced within several minutes and dissipates shortly after that. These observations confirm findings by Haxho et al. [8], in which there is a rapid process where it loses fluorescence quickly or the product of Neu-1-hydrolyzed 4-MUNANA is quickly degraded or internalized upon substrate cleavage. The readings of the fluorescence intensity were captured 3 min after the addition of 4-MUNANA sialidase substrate. Fluorometric analyses of the control (unstimulated) and CB1 agonists (AM-404, Arvanil, and Olvanil)-stimulated cell lines demonstrated the significant induction of sialidase activity in a dose-dependent response (Figure 1B).

Due to the role of NMBR in inducing MMP9 elastase activity and subsequent Neu-1 enzymatic activity, we hypothesized that NMBR inhibition using BIM-23127 would suppress Neu-1 sialidase activity in CB1-biased GPCR agonist-stimulated RawBlue cells. BIM-23127 is a specific inhibitor of NMBR. BIM-23127 (BIM23) also significantly inhibited the CB1 agonists’ activation of Neu-1 sialidase (Figure 1C). We also used an MMP-9 inhibitor (MMP-9i), a cell-permeable, potent, selective, and reversible inhibitor (IC50 5 nM). It can also inhibit MMP-1 (IC50 1.05 μM) and MMP-13 (IC50 113 nM) at higher concentrations. MMP9i, as shown in Figure 1C, significantly inhibited Neu-1 sialidase activity induced by CB1 agonists in RawBue cells. Also, oseltamivir phosphate (OP) significantly inhibited the sialidase activity induced by any of the CB1 agonists (Figure 1C). In particular, OP has been shown to explicitly target and inhibit the enzymatic activity of Neu-1 implicated with ligand-induced receptor activation [50,51].

**Figure 1 cells-13-00480-f001:**
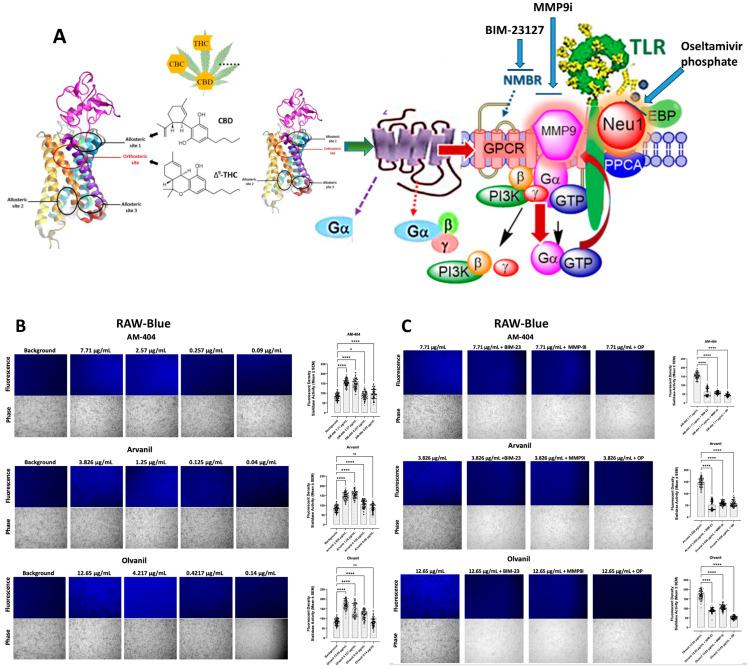
Sialidase activity is associated with CB1 GPCR agonist treatments of live RAW-blue macrophage cells. (**A**) We propose that the CB1 GPCRs heterodimerize with neuromedin B (NMBR) tethered to TLR in a multimeric receptor complex with matrix metalloproteinase-9 (MMP-9) and neuraminidase-1 (Neu1) in naïve (unstimulated) TLR-expressing cells. Here, a novel molecular signaling platform regulating the interaction and signaling mechanism between these molecules on the cell surface uncovers a functional selectivity of CB1 GPCR biased heteromers with NMBR to induce TLR activation signaling axis mediated by Neu1 sialidase activation and the modification of TLR glycosylation. This signaling platform potentiates MMP9 and Neu1 crosstalk on the cell surface, which is essential for activating TLR. Citation: Taken in part from Bunsick et al. [52]. (**B**) Cells were allowed to adhere on 12 mm circular glass slides for 24 h at 37 °C in a humidified incubator. After removing the media, 0.318 mM 4-MUNANA substrate in Tris-buffered saline pH 7.4 was added to cells alone (control background) or with either AM-404, Arvanil, and Olvanil at the indicated dosage. Fluorescent images were taken 2 min after adding substrate using epi-fluorescent microscopy (20× objective). The sialidase hydrolyzed product of 4-MUNANA (4-MU) has an emission at 450 nm (blue color) when excited at 365 nm. The mean fluorescence of 50 multi-point replicates surrounding the live cells’ periphery was calculated using Image J software, 1.5g, Java 1.8.0_345 (64-bit). The mean fluorescence ± S.E.M is represented by error bars. (**C**) Different components of the signaling paradigm were inhibited using BIM-23127, an antagonist of NMBR, MMP9i, an inhibitor of MMP-9, and oseltamivir phosphate (OP), an inhibitor of Neu-1 at the indicated predetermined concentrations. The quantified data represent two to three independent experiments displaying similar results. Statistical significance, as indicated by asterisks, was calculated with ANOVA and Fisher’s LSD uncorrected multiple comparisons post hoc test at a confidence level of 95%. ns = non-significant, **** *p* < 0.0001, * *p* < 0.05.

### 3.2. CB1 G Protein-Coupled Receptor Agonists Dose-Dependently Induce Neu1 Sialidase Activity in Pancreatic PANC-1 and Colorectal SW-620 Cancer Cell Lines

Blasco-Benito et al. [16] reported that the cannabinoid Δ9-tetrahydrocannabinol (THC) bound to the human epidermal growth factor receptor 2, forming the CB2R -HER2 complex, leading to the inactivation of HER2 through the disruption of HER2–HER2 homodimers. GPCRs have long been implicated in the transactivation of receptor tyrosine kinases (RTKs) in the absence of a respective growth factor, particularly for the receptors that bind to the epidermal growth factor, platelet-derived growth factor, fibroblast growth factor, and neurotrophins [4]. Reciprocally, growth factors binding to RTKs also utilize GPCR-signaling molecules to initiate the molecular organizational signaling platform of a novel Neu1 and MMP-9 crosstalk in alliance with RTK on the cell surface. This novel GPCR-signaling platform was identified to be critically essential for neurotrophin factor-induced TrkA [6], insulin [7,8,9,10], EGF [11], and TLR [12,13,14] receptor activation and cellular signaling.

To this end, we investigated cancer cell lines PANC-1 and SW-620 to test the hypothesis of the functional selectivity of CB1 GPCR agonist to form heteromers with NMBR to initiate the molecular organizational signaling platform of novel Neu1 and MMP-9 crosstalk in alliance with RTKs on the cell surface (Figure 2A). The data in Figure 2 revealed that CB1 GPCR agonists AM-404, Arvanil, and Olvanil used at the indicated concentrations stimulated live PANC-1 cells (Figure 2B), and SW-620 (Figure 2D) significantly and dose-dependently induced sialidase activity. There were insignificant differences in sialidase activity when the agonists exceeded their inhibitor constant KI value, signifying that the sialidase activity induced by the CB1 agonists has reached saturation.

The use of specific inhibitors with this assay also supports CB1 agonists’ involvement in this RTK signaling paradigm. One of the inhibitors used is BIM-23127, an octapeptide analog of cyclo-somatostatin with D-amino-acid substitution that functions as an antagonist of the NMBR [53]. Considering NMBR’s involvement in inducing MMP-9 elastase activity and consequent Neu-1 enzymatic activity, we hypothesized that NMBR inhibition with BIM-23127 would suppress Neu-1 sialidase activity in PANC-1 and SW-620 cells stimulated with CB1 agonists (at the most potent concentrations), AM-404, Arvanil, and Olvanil. We found that the NMBR antagonist BIM-23127 inhibits Neu-1 sialidase activity stimulated by the CB1 agonists in all three cell lines (Figure 2B,D). Interestingly, we observed a low fluorescence inhibition following treatment with BIM-23 compared to the other two agonists, suggesting that it may be a weaker inhibitor. It is possible that the concentration of this agonist needs to be increased to have a more substantial inhibitory effect, or that the presence of agonists is stronger than the inhibitor.

Our findings also confirm that MMP-9i significantly inhibits the sialidase activity induced by CB1 agonists at the saturation concentration compared to treatment with the agonists alone (Figure 2B,D). Also, we confirm that OP exerts a significant impact in suppressing sialidase activity in CB1-stimulated live cells when compared to cells treated without OP despite treatment with high concentrations of CB1 agonists (Figure 2B,D). These results further substantiate that the findings are due to the CB1-induced stimulation of the signaling axis and consequent activation of the Neu-1 sialidase assay. Notably, the suppression of MMP-9 demonstrates a more pronounced impact on diminishing sialidase activity when contrasted with BIM-23127, albeit with a lower effect than observed with OP. This phenomenon may be elucidated through the consequential signaling of the axis, wherein MMP-9 more directly interacts with Neu-1, activating the sialidase domain.

### 3.3. CB1 Receptor Co-Localizes with Neu1 on the Cell Surface of Naïve Unstimulated RAW-Blue Macrophages, PANC-1, and SW-620 Cells

Emerging research regarding the CB1 receptors reveals that CB1 GPCR can interact with EGF receptors [54], Trk receptors [55], and TLRs [56]. We have reported that these RTK and TLR receptors are regulated by the molecular organizational signaling platform of Neu1 and MMP-9 crosstalk in alliance with GPCR on the cell surface with neurotrophin factor-induced TrkA [6], insulin [7,8,9,10], epidermal growth factor (EGF) [11], and Toll-like (TLR) [12,13,14] for receptor activation and cellular signaling. Here, we investigated whether CB1 forms a complex with the signaling platform of Neu1 and MMP-9 crosstalk in alliance with NMBR GPCR on the cell surface. As depicted in Figure 3, CB1 receptors co-localize with Neu-1 in naïve RAW-Blue macrophages, PANC-1, and SW-620 cells. Here, we used the Pearson correlation coefficient, which measures the linear association between two variables. The Pearson correlation coefficient for CB1-Neu-1 co-localization was 0.5917 in RAW-Blue cells (A), 0.6467 in PANC-1 cells (B), and 0.6455 in SW-620 cells (C), affirming CB1-Neu1 receptor proximity.

Collectively, these findings provide statistically significant support for the participation of the CB1 receptors involved in the proposed biased GPCR signaling paradigm. Interestingly, these findings provide novel insight into how CB1 receptors co-localizing with Neu-1 may be poised to assume biased active states that are linked to RTK and TLR with subsequent NF-kB activation. Regulation of these CB1 GPCR active states may be mediated by endogenous CB1 agonists binding to either sites or spatially different allosteric sites [57], leading to distinct functional selectivity or ‘biased agonism.’ Furthermore, in the context of ligand-binding receptor and ligand-stimulated receptor signaling kinetics, the results provide new insights into the intricate interaction between G proteins and CB1 receptors at the resting state. This approach underscores the concept that biased agonism is influenced by G protein composition, mainly through distinct receptor G protein differentiation [57,58].

### 3.4. Synthetic CB1 Cannabinoids AM-404, Aravnil, and Olvanil Marginally Reduce the Expression of E-Cadherin in SW-620 Colorectal Cancer Cells

E-cadherin is a cell–cell adhesion protein with a critical role in epithelial cell behavior and cancer suppression [59]. E-cadherin is used in the diagnosis and prognosis of epithelial cancers as reduced expression is associated with tumors undergoing EMT [60]. Moreover, other studies have shown that the increased expression of Neu-1 promotes EMT through the reduction in E-cadherin markers [61,62]. To determine the influence of the CB1 agonists on the expression of E-cadherin, we examined the presence and levels of E-cadherin as a marker of EMT in SW-620 colorectal cancer cells. A decrease in E-cadherin following CB1 agonist treatment would not only indicate cancerous cells undergoing EMT but also indicate that CB1 treatment augments the process. As depicted in Figure 4, SW-620 cells, there was no statistically significant reduction in E-cadherin on SW-620 cells following CB1 agonist treatment when compared to the control group. However, there is a slight reduction in E-cadherin expression following the treatment. There are two potential reasons for these results. Firstly, the CB1 agonists may have no significant impact on E-cadherin levels. Secondly, given that SW-620 cells have already undergone the EMT process for invasiveness, it is possible that there has already been a significant reduction in E-cadherin and that further treatment with the agonists has no further effect on the marker. Despite this, treatment with the CB1 agonists did not increase E-cadherin expression, indicating that these synthetic cannabinoids may not have the tumor suppression function that was once thought. While the majority of research indicates that a decline in E-cadherin expression is a characteristic feature of EMT, the recent literature contends that the absence of E-cadherin is neither causal nor necessary for EMT [63,64]. As such, the study findings are supported by the literature, providing further insight into the specific effects of CB1 agonists on EMT marker expression.

### 3.5. Synthetic CB1 Cannabinoids AM-404, Arvanil, and Olvanil Significantly Upregulate the Expression of Vimentin in SW-620 Cells

Vimentin is a structural protein with a role in regulating the mechanical and motile properties of cells [65]. Clinical evidence supports a causal link between vimentin expression and the metastatic process, whereby vimentin is highly expressed in primary and metastatic tumors [65,66]. Vimentin was selected as the marker for the SW-620 cells. Previous research indicates that N-cadherin is not present in these cells, suggesting that they are not mesenchymal as they have already undergone EMT [67]. Our previous results showed a link between CB1 and Neu-1, and given Neu-1’s influence on EMT markers [68], we investigated if SW-620 cells treated with CB1 agonists would increase the expression of vimentin. An increase in vimentin expression would be an indication that CB1 agonists enhance the metastatic progression of the SW-620 colorectal cancer cells. A pronounced upregulation of vimentin expression in response to CB1 agonists in SW-620 cells when compared to the untreated control was noted (Figure 5). Noteworthy, although all three agonists upregulated vimentin expression, Olvanil had a weaker and non-significant effect on vimentin expression, respectively (Figure 5). These results, however, can likely be attributed to the strength of the agonists as Olvanil is a weaker CB1 agonist, as represented by its KI value when compared to the other two CB1 agonists. The fact that agonist strength impacted the expression of vimentin suggests that there is a kinetic component within the CB1 receptor that needs to be studied further. Ultimately, these observations contribute to the previous findings, supporting the proposed consequences of CB1-GPCR-Neu1-MMP-9 axis stimulation in cancer metastasis.

### 3.6. CB1 Agonists, AM-404, Arvanil, and Olvanil Induce Upregulation of NF-kB

To better study the mechanism of the induction of the EMT induced by CB1 agonists, AM-404, Arvanil, and Olvanil, we investigated the effect of CB1 agonists for their ability to upregulate nuclear factor-κB (NF-κB) when compared to positive control lipopolysaccharide (LPS). NF-kB represents a family of transcription response profiles. Here, we used RAW-Blue™ cells (mouse macrophage reporter cell line). These cells stably express a secreted embryonic alkaline phosphatase (SEAP) gene inducible by NF-kB and activator protein-1 (AP-1) transcription factors [33]. AP-1, a transcription factor, regulates the expression of genes responding to various stimuli, including cytokines, growth factors, stress, and bacterial and viral infections. Upon stimulation, RAW-Blue™ cells activate NF-kB and AP-1, leading to the secretion of SEAP, which is detectable and measurable when using QUANTI-Blue™, a SEAP detection medium. Since Neu-1 activity is associated with GPCR-signaling and MMP-9 activation in live TLR-expressing macrophage cells [13], we asked if CB1 GPCR agonists would directly induce NF-kB in the absence of any TLR-specific ligand. The data in Figure 6 are consistent with this hypothesis. Here, GPCR agonists AM-404, Arvanil, and Olvanil heterodimerizing with neuromedin B receptor (NMBR) tethered to TLR receptors induced NFκB-dependent secretory alkaline phosphatase (SEAP) activity in live RawBlue macrophage cells compared to LPS in a dose-dependent manner. Collectively, the additional intracellular and cell surface co-localization of CB1, Neu1, and MMP-9 validated the predicted crosstalk between the CB1-NMBR–MMP-9–Neu1 partite complex tethered to TLR receptors.

## 4. Discussion

Here, we provide evidence that synthetic CB1 agonists AM-404, Arvanil, and Olvanil each significantly and dose-dependently induce sialidase activity in live RAW-Blue murine macrophages, PANC-1, and SW-620 cells in vitro. To support our hypothesis that these agonists induce Neu-1 specifically, we used three specific Neu-1 inhibitors, BIM-23127, MMP-9i, and OP, and found sialidase activity to be blocked. Moreover, these findings support our proposed signaling paradigm, where CB1 signaling induces Neu-1 sialidase and MMP-9 crosstalk. To confirm these results, a co-localization assay was used to determine CB1 GPCR’s proximity to Neu-1 on the cell surface. The results firmly established that CB1 GPCR and Neu-1 are in complex with each other, in support of our hypothesis. Previous studies have highlighted Neu-1’s role in tumorigenesis [68], inflammation [69], and insulin resistance [70], all of which are hallmarks of cancer progression and metastasis. We then hypothesized that CB1 agonists increase the expression of EMT markers, including vimentin and N-cadherin. Although one agonist, Olvanil, did not significantly increase vimentin expression, the other agonists increased vimentin expression in SW-620 colorectal cancer cells. These agonists also marginally decreased E-cadherin expression, suggesting these agonists have the potential to play a role in EMT if present.

The evidence supporting the involvement of CB1 in the Neu1-NMBR-MMP9 biased signaling paradigm is twofold. Firstly, existing research supports the existence of the CB1-NMBR heterodimerization. CB1 is known to heterodimerize with a variety of GPCRs, and as research in this field has progressed, novel CB1 heterodimers have continued to emerge [71,72,73]. Furthermore, De Petrocellis et al. and other researchers demonstrated that cannabinoids could alter bombesin (BRS-3) receptor signaling, a receptor with significant structural similarity to NMBR [74,75,76]. Specifically, the CB1 agonist arachidonoyl-chloro-ethanolamine (ACEA) strongly inhibited bombesin-induced Ca^2+^ elevation, while a different cannabinoid, HU210, moderately inhibited bombesin-induced Ca^2+^ elevation [74]. These results demonstrated crosstalk between the bombesin GPCR and CB1 receptors, which could be mediated by heterodimerization and the potential for ligand-directed functional selectivity. It is possible that certain cannabinoids more strongly induce BRS3-CB1 heterodimerization, leading to differences in G protein coupling, which ultimately produce biased effects on downstream Ca^2+^ concentrations. Overall, as NMBR and BRS-3 both belong to the bombesin-like receptor subfamily and share significant structural similarities, these findings from De Petrocellis and colleagues provide a rationale for the functional and structural interactions between CB1 and NMBR [75,76].

The second central précis supporting the involvement of CB1 in the NMBR-MMP9-Neu1 biased signaling paradigm is that cannabinoids are known to transactivate various glycosylated receptors, including the vascular endothelial growth factor receptor (Flk-1 VEGFR), epidermal growth factor receptor, and insulin-like growth factor 1 receptor [77]. Furthermore, a study by Haxho et al. [8] reported that the GPCR agonists bradykinin, angiotensin I, and angiotensin II induce Neu-1 sialidase activity in a dose-dependent manner in HTC-IR cells. This study suggested that GPCR agonism and IR glycosylation enable Neu-1-mediated desialylation of the IRβ subunits to transactivate the IR. Notably, a report by Rozenfeld et al. [15] suggested a link between the CB1 and angiotensin receptors, supporting the concept of the CB1 receptor’s involvement in IR transactivation by the Neu1-NMBR-MMP9 signaling axis. This study found that the Gα coupled type 1 cannabinoid receptor and the Gα/q coupled AT1R (angiotensin 1 receptor) functionally interacted, resulting in AT1R signaling and the coupling of AT1R to several G proteins. The study also found that CB1 upregulation increases AT1R-CB1 heteromers, enhancing angiotensin-mediated signaling. When inhibiting CB1 activity, angiotensin II-mediated mitogenic signaling and gene expression decreased. The evidence suggests a link between the two receptors, indicating that they may have similar properties regarding GPCR signaling and biased agonism. It is noteworthy that the results of this study found that CB1R expression regulates another GPCR as it increases the relevancy of CB1R upregulation in chronic disease states, suggesting that the CB1R may have interactions with other GPCRs that remain to be elucidated. Notably, there is evidence to suggest that CB1 heteromers may have a similar effect on the activation of the Neu1-NMBR-MMP9 signaling axis. Further investigations into the involvement of this signaling axis in mediating CB1-biased effects may have important implications for pharmaceutical development and uncover new mechanisms regarding insulin signaling and cancer metastasis.

It is not surprising that CB1 is in complex with Neu-1 as both have interactions with TLR4 [56,69,78], EGFR [11,54], Trks [6,79], and IR [7,8,9]. There is no debate that the increased expression of Neu-1 and MMP-9 has an impact on cancer progression through the modulation of inflammation, tumorigenesis, and insulin receptor signaling [52,68,80]. However, there is conflicting evidence on CB1 agonism and cancer metastasis. Alternatively, some studies support the notion that CB1 agonism promotes cancer metastasis [52,81]. With findings that contradict one another, it may be challenging to elucidate the effectiveness by which endocannabinoids modulate cancer progression. Despite this, the lack of conclusive findings may pave the way to a more comprehensive understanding of functional selectivity in CB1 signaling. Many of these studies utilized THC or CBD. THC displays a moderate affinity for CB1 receptors, acting as a partial agonist, and CBD has a lower affinity for CB1 [82]. The weaker agonists may be unable to evoke a strong and consistent CB1 signal to upregulate cancer metastasis markers. Our results show that while the more potent CB1 agonists, AM-404 and Arvanil, increased vimentin expression, whereas the weaker agonist, Olvanil, did not. Moreover, the results suggest that strong CB1 signaling may result in biased agonism where consistent signaling of the CB1 receptor may upregulate GPCRs, including Neu-1, which have a malignant effect on cancer.

The results presented in this paper provide an alternative explanation that follows the ideology of the GPCR functional selectivity program. Orthosteric, allosteric, and biased agonisms all contribute to the concept of functional selectivity. A review by Morales et al. [83,84] highlights the structural features of GPCRs and how various intracellular conformations result in specific coupling to effector proteins, impacting the signaling pathway being activated. For example, activation of the transmembrane helix 6 enables Gα protein insertion into the GPCR, whereas transmembrane helix 7 favors β-arrestin coupling. However, a third conformation is possible, as described by Haxho et al. [8], where the ligand heterodimerizes with NMBR before activating the Gα protein pathway.

Given that CB1 is a GPCR, it is likely that functional selectivity occurs where CB1 agonists may bind to the receptor, resulting in the activation of any three of these GPCR pathways. Although CB1 agonists all bind and activate the CB1 receptor, their unique conformation and binding to the CB1 receptor may likely produce a different response compared to another agonist with a slightly altered conformation. It is also possible that the presence of multiple ligands affects CB1 signaling. A perfect example was demonstrated by Laprairie et al. [85], who found that, on their own, THC and CBD activated the beta-arrestin pathway in wild-type and Huntington-expressing cells. However, the presence of both ligands favored the alpha–gamma signaling pathway. The molecular understanding of the biochemistry behind the binding of these agonists to these receptors may aid in the understanding of how different agonists evoke unique responses in the CB1 signaling pathway. The interactions of these ligands on their own and in tandem are critical in our understanding of how cannabis and their CB1 receptors are to be considered for future therapies.

## 5. Conclusions

Functional selectivity provides an exciting new direction to maximize the therapeutic potential of GPCR-targeting drugs while minimizing adverse effects. However, our understanding of how cannabinoid ligands produce biased effects is still in its infancy. For this concept of biased agonism to contribute to expanding cannabinoid-based pharmaceuticals in healthcare, we must ascertain a more comprehensive understanding of the specific mechanisms of CB1 receptor functional selectivity. As presented here, investigations into the Neu1-NMBR-MMP9 signaling axis may provide unprecedented insights into these mechanisms.

While previous studies confirm that cannabinoids can induce functional selectivity at the CB1 receptor, the mechanisms underlying this phenomenon remain poorly understood. Nevertheless, given the importance of dimerization at this receptor subtype, mechanisms that involve the formation of CB1 dimers and oligomers are likely to be involved. For example, one oligomer-based signaling axis shown to mediate biased agonism at various GPCRs consists of the oligomerization of the GPCR with the NMBR receptor and a glycosylated receptor [52,86,87,88]. The dimerization of the GPCR with NMBR mediates crosstalk between the neuraminidase-1 (Neu-1) and matrix metalloproteinase-9 (MMP-9) enzymes, ultimately leading to the transactivation of glycosylated receptors and the biased activation of downstream signaling pathways. While this signaling axis has yet to be studied in the context of the CB1 receptor, compelling evidence exists supporting its ability to contribute to functional selectivity at the cannabinoid receptor.

This interesting imperative is underscored as emerging evidence has implicated the involvement of CB1 receptors in a biased GPCR signaling platform involving the dimerization of the GPCR with NMBR and subsequent crosstalk between the neuraminidase-1 and matrix metalloproteinase-9 enzymes, as reviewed elsewhere [9,52]. As depicted in Figure 7, CB1 agonists may also induce epigenetic modifications by inducing the biased CB1-NMBR-Neu1-MMP9 signaling paradigm and indirectly stimulating the TLR through a process known as transactivation [8,52,86,89]. Subsequent activation of these glycosylated receptors facilitates the initiating and translocation of the NF-kB pathway into the nucleus, enabling the epigenetic reprogramming of gene expression [89].

This concept elegantly substantiates the conceptual framework proposing the activation of the biased CB1 GPCR-Neu1-MMP9 signaling platform by a select group of cannabinoids activating the NF-kB pathway and subsequent epigenetic remodeling, implicated in enhanced EMT markers and metastasis [88]. Although the role of the biased CB1 GPCR-Neu1-MMP9 signaling platform and the associated activation of NF-kB has yet to be studied concerning cancer epigenetics and metastasis, NF-kB presents a significant potential as a mediator of the epigenetic effects induced by cannabis [91]. Notably, studies have implicated CBD in suppressing the NF-kB pathway, strongly indicating its anti-proliferative and anti-inflammatory activities [92]. Conversely, THC demonstrates a propensity to induce the NF-kB pathways, which in turn assumes a multifaceted role as a consequential epigenetic regulator of diverse gene expression in malignant tumors, thereby implicating its involvement in EMT [93,94]. Nonetheless, the mechanisms underlying the functional selectivity of NF-kB signaling in the context of CB receptors remain to be investigated.

## Figures and Tables

**Figure 2 cells-13-00480-f002:**
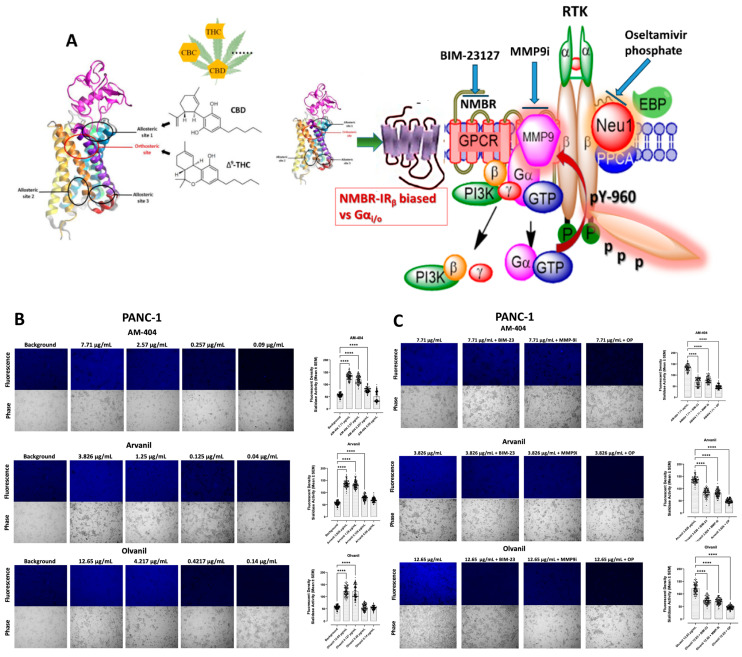
(**A**) We propose that the CB1 GPCRs heterodimerize with neuromedin B (NMBR) tethered to RTK in a multimeric receptor complex with matrix metalloproteinase-9 (MMP-9) and neuraminidase-1 (Neu1) in naïve (unstimulated) cancer cells. Citation: Taken in part from Bunsick et al. [52] and Haxho et al. [9]. Sialidase activity of live PANC-1 (**B**,**C**) and SW-620 (**D**,**E**) in response to AM-404, Arvanil, and Olvanil. CB1 agonists, AM-404, Arvanil, and Olvanil, significantly induce Neu-1 sialidase activity in a dose-dependent fashion compared to media control in live PANC-1 (**B**) and SW-620 (**D**). In the three cell lines, the saturation and KI concentrations for each agonist produced a more significant impact on sialidase activity compared to the latter two concentrations, which were more representative of the control. The sialidase hydrolyzed product of 4-MUNANA (4-MU) has an emission at 450 nm (blue color) when excited at 365 nm. The mean fluorescence of 50 multi-point replicates surrounding the cell’s periphery was calculated using Image J software. The mean fluorescence ± S.E.M is represented by error bars. (**C**,**E**) Different components of the signaling paradigm were inhibited using BIM-23127, an antagonist of NMBR, MMP9i, an inhibitor of MMP-9, and oseltamivir phosphate (OP), an inhibitor of Neu-1 at the indicated predetermined concentrations. The quantified data represent two to three independent experiments displaying similar results. Statistical significance, as indicated by asterisks, was calculated with ANOVA and Fisher’s uncorrected LSD multiple comparisons post hoc test at a confidence level of 95%. ns = non-significant, **** *p* < 0.0001, ** *p* < 0.01.

**Figure 3 cells-13-00480-f003:**
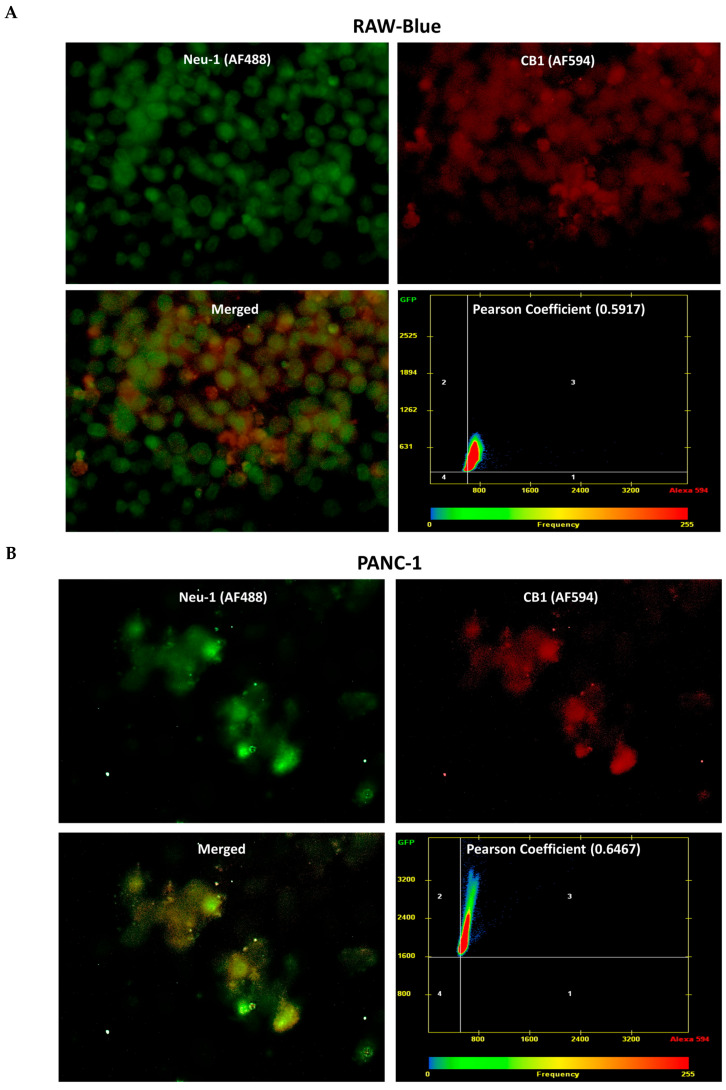
Co-localization of cannabinoid type 1 (CB1) receptor with Neu-1. CB1 receptors co-localized with Neu-1 in naïve, unstimulated RAW-Blue (**A**), PANC-1 (**B**), and SW-620 (**C**) cells. For analysis of co-localization, cells were fixed, permeabilized, blocked, and immunostained with anti-CB1 conjugated with AlexaFluor 594 (red) and anti-Neu-1 conjugated with AlexaFluor 488 (green). Co-localization was quantified using AxioVision imaging software to compute the Pearson correlation coefficient, measuring the linear association between two variables (40× objective). The Pearson correlation coefficient for co-localization between CB1 receptors and Neu-1 was 0.5917 in RAW-Blue (**A**), 0.6467 in PANC-1 (**B**), and 0.6455 in SW-620 (**C**), affirming CB1-Neu1 receptor proximity.

**Figure 4 cells-13-00480-f004:**
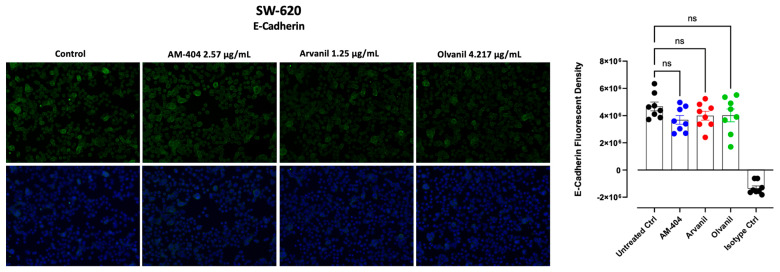
Immunofluorescence analysis of E-cadherin expression in response to CB1-agonists. CB1-agonists, AM-404, Arvanil, and Olvanil notably reduced the expression of the epithelial–mesenchymal transition (EMT), E-cadherin, in SW-620 cells. Cells were fixed, permeabilized, blocked, and immunostained with primary mouse monoclonal IgG antibodies for E-cadherin, followed by goat anti-mouse AlexaFluor 488 (green) secondary antibodies. As an isotype control, we use normal mouse IgG. The fluorescence was calculated using Corel Photo-Paint, with the average of eight points subtracted by the background fluorescence and multiplied by the pixel density (20× objective). Statistical significance was calculated with ANOVA and Fisher’s uncorrected LSD multiple comparisons post hoc test at a confidence level of 95%. ns = non-significant.

**Figure 5 cells-13-00480-f005:**
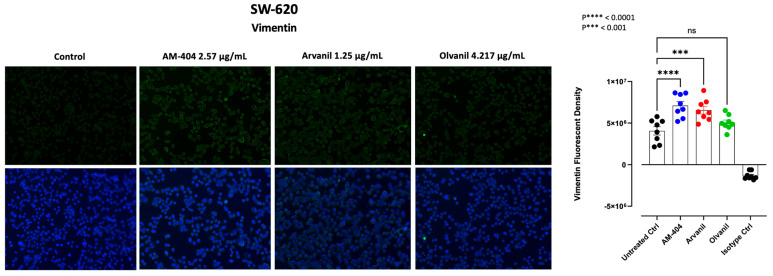
Immunofluorescence analysis of vimentin expression in response to CB1-agonists. CB1-agonists, AM-404, Arvanil, and Olvanil, significantly increased the expression of the epithelial–mesenchymal transition (EMT), vimentin, in SW-620 cells. Cells were fixed, permeabilized, blocked, and immunostained with primary mouse monoclonal IgG antibodies for vimentin, followed by goat anti-mouse AlexaFluor 488 (green) secondary antibodies. As an isotype control, we use normal mouse IgG. The fluorescence was calculated using Corel Photo-Paint, with the average of eight points subtracted by the background fluorescence and multiplied by the pixel density (20× objective). Statistical significance, as indicated by asterisks, was calculated with ANOVA and Fisher’s uncorrected LSD multiple comparisons post hoc test at a confidence level of 95%. ns = non-significant, **** *p* < 0.0001, *** *p* < 0.001.

**Figure 6 cells-13-00480-f006:**
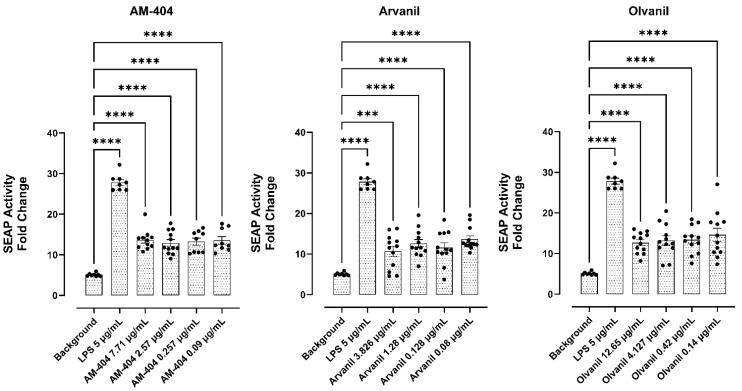
The cells stably express a secreted embryonic alkaline phosphatase (SEAP) gene inducible by NF-kB and AP-1 transcription factors. Upon stimulation, RAW-Blue™ cells activate NF-kB and AP-1, leading to the secretion of SEAP, which is detectable and measurable when using QUANTI-Blue™, a SEAP detection medium (Invivogen). Quantitative spectrophotometry analysis of the effect of LPS and AM-404-, Arvanil-, and Olvanil-induced SEAP activity in the culture medium. The measurement of the relative SEAP activity was calculated as fold change in each compound (SEAP activity in medium from treated cells minus no cell background over SEAP activity in medium from untreated cells minus background). Results are the means of three separate experiments. Statistical significance, as indicated by asterisks, was calculated with ANOVA and Fisher’s uncorrected LSD multiple comparisons post hoc test at a confidence level of 95%.**** *p* < 0.0001, *** *p* < 0.001.

**Figure 7 cells-13-00480-f007:**
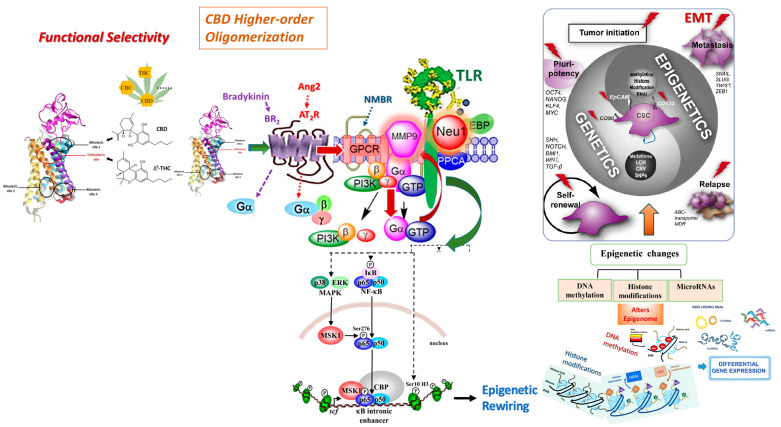
Cannabinoid type 1 (CB1) receptor participates in a multimeric receptor complex with neuromedin B receptor (NMBR), neuraminidase 1 (Neu-1), and Toll-like receptor (TLR). Here, the biased CB1 GPCR platform potentiates Neu-1 and MMP-9 cell surface crosstalk, mediating TLR glycosylation modification and transactivation and subsequent NF-kB-induced epigenetic rewiring. ∆^9^-tetrahydrocannabinol (THC), an orthosteric ligand, and cannabidiol (CBD), an allosteric ligand, alongside the N-terminus and location of the orthosteric and allosteric binding sites, are illustrated concerning CB1 cannabinoid receptors. Notes: CB1 agonists stimulation of the biased CB1 GPCR induces receptor heterodimerization with NMBR, wherein NMBR-induced activation of MMP-9 allows the endopeptidase to cleave the elastin binding protein and expose the catalytic sialidase domain of Neu-1. The sialidase domain of Neu-1 hydrolyzes alpha-2,3-sialic acid from the glycosylated receptor, TLR, reducing the steric hindrance, which facilitates TLR dimerization, activation, and cellular signaling. The resultant downstream signaling mediates the phosphorylation of the IkB subunit, which facilitates the translocation of NF-kB to the nucleus, enabling epigenetic modulation of gene expression [8,89]. Citation: Reprinted/Adapted with permission, Bunsick et al. [52], Qorri et al. [86], Jakowiecki et al. [87], Reber et al. [88], and Marquardt et al. [90] Licensee MDPI, Basel, Switzerland. These articles are openaccess articles distributed under the terms and conditions of the Creative Commons Attribution (CCBY) license (http://creativecommons.org/licenses/by/4.0/ (23 April 2021), which permits unrestricted use, distribution, and reproduction in any medium, provided the original author and source are properly credited.

## Data Availability

All data needed to evaluate the paper’s conclusions are present. The preclinical datasets generated and analyzed during the current study are not publicly available but are available from the corresponding author upon reasonable request. The data will be provided following the review and approval of a research proposal, Statistical Analysis Plan, and execution of a Data Sharing Agreement. The data will be accessible for twelve months for approved requests, considering possible extensions; contact szewczuk@queensu.ca for more information on the process or to submit a request.

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
