# Peer review of "Functional Selectivity of Cannabinoid Type 1 G Protein-Coupled Receptor Agonists in Transactivating Glycosylated Receptors on Cancer Cells to Induce Epithelial–Mesenchymal Transition Metastatic Phenotype"

_cells, 2024, doi:10.3390/cells13060480_

Round 1
Reviewer 1 Report (New Reviewer)
Comments and Suggestions for Authors
I believe the manuscript can potentially contribute significantly to cancer cell metastasis. However, the authors must address the following comments to consider the manuscript for publication.
-As an original manuscript, the introduction section is so gloomy for readers that it needs to be written in a better way to be interesting and understandable for readers.
-In the introduction, I cannot understand the reasoning of many sections, and there is no reasonable arrangement.
-The manuscript's goal(s) are not yet ready for readers to find them.
-The introduction is like a review article and needs significant modifications from the original one.
-Methods need a section on the strategy of treatment (time and dosage) and the reason for them (AM-404, Arvanil, and Olvanil).
-Some methods need references.
-In section 5.1, the cell line type is enough; we don’t need more.
-The result parts were explained like methods in the beginning, and I also don’t understand the reference in the result section.
-The discussion section needs major modification.
Author Response
I believe the manuscript can potentially contribute significantly to cancer cell metastasis. However, the authors must address the following comments to consider the manuscript for publication.
-As an original manuscript, the introduction section is so gloomy for readers that it needs to be written in a better way to be interesting and understandable for readers. -In the introduction, I cannot understand the reasoning of many sections, and there is no reasonable arrangement. -The introduction is like a review article and needs significant modifications from the original one.
Author response: Thank you for this comment and suggestion. Originally, the manuscript was written as a review article, but one reviewer want data to substantiate the concepts of the paper. We have entirely written the introduction focusing on the main topic of CB1 biased functional selectivity. Here, we provided evidence for CB1 agonists molecular mechanisms involved in this novel cross-communication between GPCRs and RTKs and TLRs. Rozenfeld et al. [15] reported a AT1R–cannabinoid CB1R heterodimerization complex, revealing a new mechanistic action for the pathogenic properties of angiotensin-2. Amidst the interests in cannabis usage, there is an increase in research on the impacts of the endocannabinoids (ECS), cannabinoids, and associated CB1/CB2 receptors on tumour development and metastasis [28]. Of the existing literature, evidence has revealed that cannabinoids can impact markers of epithelial-mesenchymal transition (EMT), one of the hallmarks of cancer metastasis [29,30] [31] [32]. However, there exists substantial variability in the responses of cancer cells to cannabinoids in the context of invasiveness and metastatic capacities. This is a brief of the outline of the revised introduction which leads to the rationale of our proposed research.
The manuscript's goal(s) are not yet ready for readers to find them.
Author response: Thank you for this comment and suggestion. The manuscript goals and rationale are now clearly outlined in the revised introduction.
-Methods need a section on the strategy of treatment (time and dosage) and the reason for them (AM-404, Arvanil, and Olvanil).
Author response: Thank you for this comment and suggestion. We have added a new section 2.3 CB1 Agonist Treatment Protocol (time and dosage). The agonists AM-404, Arvanil, and Olvanil were chosen based on their interactions with the CB1 receptor and binding affinity. AM-404 is an indirect agonist of the CB1 receptor by binding to the fatty acid amide hydrolase (FAAH) enzyme, preventing anandamide breakdown [34,35]. AM-404 has a Ki value of ~6.5 µM (2.57 µg/mL) [36-39]. Aravnil is a potent direct agonist of the CB1 receptor and prevents anandamide breakdown. The agonist has a Ki value of ~2.9 µM (1.27 µg/mL) [34,39-41]. Olvanil is also a direct agonist of the CB1 receptor and prevents anandamide breakdown. Olvanil is a weaker agonist with a Ki value of ~10.1 µM (4.21 µg/mL) [39,42,43]. These values were used to create a dose-response curve where the Ki value is a 1:3 dilution of the saturation dose. A 1:10 and 1:30 dilution was made from the saturation dose. The cells were treated with the cannabinoid for 24 hours as previously reported by others [44-46].
-Some methods need references.
Author response: Thank you for this comment and suggestion. We have added citations where appropriate.
-In section 5.1, the cell line type is enough; we don’t need more.
Author response: Thank you for this comment and suggestion. We have revised accordingly: Three cell lines were used in these studies, PANC-1 (ATCC® CRL-1469™), RAW-Blue macrophages (InvivoGen), and SW-620 (ATCC® CCL-227™) cells.
-The result parts were explained like methods in the beginning, and I also don’t understand the reference in the result section.
Author response: Thank you for this comment and suggestion. The results section is also entirely rewritten focusing on the rationale for each of the Results sections.
-The discussion section needs major modification.
Author response: Thank you for this comment and suggestion. The discussion section is also rewritten explaining the results of the study and its significance. We have included a revise conclusion with an updated graphical abstract focusing on the significance of the research study.
Reviewer 2 Report (New Reviewer)
Comments and Suggestions for Authors
The article submitted for expert appraisal is a study of the functional selectivity of agonists of the cannabinoid GPCR CB1 transactivates glycosylated receptors in cancer cells to induce the EMT metastatic phenotype.
The comprehensive study leads to a large number of results. These results are scarcely discussed; the discussion section needs to be improved. The authors need to show how their results are new in relation to what is already in the bibliography.
Similarly, the introduction needs to be improved, so as to properly introduce the objective of their study and specify their method.
The presentation of results by figure is clear, although Figure 1 should be split into two or improved. Figure titles are far too long, and in particular the note to Figure 1 should be simplified or included in the body of the text.
At the end of the article, there are two "Author Contributions" sections?
Author Response
The article submitted for expert appraisal is a study of the functional selectivity of agonists of the cannabinoid GPCR CB1 transactivates glycosylated receptors in cancer cells to induce the EMT metastatic phenotype.
The comprehensive study leads to a large number of results. These results are scarcely discussed; the discussion section needs to be improved.
Author response: Thank you for this comment and suggestion. Originally, the manuscript was written as a review article, but one reviewer want data to substantiate the concepts of the paper. We have entirely written the introduction focusing on the main topic of CB1 biased functional selectivity. Here, we provided evidence for CB1 agonists molecular mechanisms involved in this novel cross-communication between GPCRs and RTKs and TLRs. Rozenfeld et al. [15] reported a AT1R–cannabinoid CB1R heterodimerization complex, revealing a new mechanistic action for the pathogenic properties of angiotensin-2. Amidst the interests in cannabis usage, there is an increase in research on the impacts of the endocannabinoids (ECS), cannabinoids, and associated CB1/CB2 receptors on tumour development and metastasis [28]. Of the existing literature, evidence has revealed that cannabinoids can impact markers of epithelial-mesenchymal transition (EMT), one of the hallmarks of cancer metastasis [29,30] [31] [32]. However, there exists substantial variability in the responses of cancer cells to cannabinoids in the context of invasiveness and metastatic capacities. This is a brief of the outline of the revised introduction which leads to the rationale of our proposed research.
the discussion section needs to be improved.
Author response: Thank you for this comment and suggestion. The discussion section is also rewritten explaining the results of the study and its significance. We have included a revise conclusion with an updated graphical abstract focusing on the significance of the research study.
The authors need to show how their results are new in relation to what is already in the bibliography.
Author response: Thank you for this comment and suggestion. The introduction and discussion sections are rewritten explaining the results of the study and its significance. We have included a revise conclusion with an updated graphical abstract focusing on the significance of the research study, and how the results are novel in relation to what is already in the bibliography.
Similarly, the introduction needs to be improved, so as to properly introduce the objective of their study and specify their method.
Author response: Thank you for this comment and suggestion. DONE
The presentation of results by figure is clear, although Figure 1 should be split into two or improved. Figure titles are far too long, and in particular the note to Figure 1 should be simplified or included in the body of the text.
Author response: Thank you for this comment and suggestion. We have revised the organization of the figures to focus on the main theme of the study.
At the end of the article, there are two "Author Contributions" sections?
Author response: Thank you for this comment and suggestion. We have changed one of them to “Author Acknowledgements.”
Round 2
Reviewer 1 Report (New Reviewer)
Comments and Suggestions for Authors
The manuscript is acceptable for publication.
Reviewer 2 Report (New Reviewer)
Comments and Suggestions for Authors
The authors have responded fully to suggestions and comments, so the article can now be published.
This manuscript is a resubmission of an earlier submission. The following is a list of the peer review reports and author responses from that submission.
Round 1
Reviewer 1 Report
Comments and Suggestions for Authors
In this manuscript, the author reviews the literature on the role of biased GPCR agonisms and their relevance to the CB1 receptor. This is an essential and debated issue in pharmacological research; therefore, this review is timely. This review is clearly written and provides an important contribution to the field.
Minor point
1. The lettering in Figure 1 is too small; please increase the size.
2. Abbreviation of PPCA is lacking on the test and the figure.
3. Please add the abbreviation in the legend of Fig 1.
Author Response
REVIEWER #1
Comments and Suggestions for Authors
In this manuscript, the author reviews the literature on the role of biased GPCR agonisms and their relevance to the CB1 receptor. This is an essential and debated issue in pharmacological research; therefore, this review is timely. This review is clearly written and provides an important contribution to the field.
Minor point
- The lettering in Figure 1 is too small; please increase the size.
- Abbreviation of PPCA is lacking on the test and the figure.
- Please add the abbreviation in the legend of Fig 1.
Author response: Thank you for the comment. We have increased the letters in Figure 1 as suggested and included a description of PPCA and EBP in the figure legend. We have added an Abbreviation to the figure legend.
Reviewer 2 Report
Comments and Suggestions for Authors
In this review Matsukobo et al. describe the importance of biased agonism and allosteric modulation of GPCRs in activating distinct intracellular signaling pathways. They highlight primarily the modifications induced by cannabinoids on diverse receptors and pathways and discuss the relevance of these mechanisms in the development of novel therapeutic strategies. The review is very original and interesting for a broad audience but the authors should address the following points:
1. The authors must provide some examples of how biased agonism, allosteric modulation and transactivation of glycosylated receptors “in health and disease” (as the Title dictates) by cannabinoids may lead to an “exciting new direction to maximize the therapeutic potential of cannabinoids and GPCR-targeting drugs while minimizing adverse effects”.
2. If the terms “biased agonism” and “functionally selectivity” are synonyms, they should be both introduced together as such in the Abstract and in the title of section 1.1
3. The legend to Figure 1 is quite difficult to understand. Some sentences appear to be truncated, and some others (in the final part) appear to be repeated. The CB1 receptor should be labeled in the figure.
Comments on the Quality of English LanguageThe English language is fine but some minor errors should be corrected
Author Response
REVIEWER #2
Comments and Suggestions for Authors
While the topic of this mini-review is interesting, basic research supporting the transactivation of glycoslyated receptors by CB1 receptors is very limited. This review is therefore certainly premature as more basic research needs to be performed. There is a danger that it rather misleads the research community. The interaction of CB1 receptors with NMBR is not sufficiently substantiated by actual data. The model shown in Figure 1 is speculative and would serve as the starting point to perform more research. It is problematic when early ideas are published as scientific concepts.
Author response: Thank you for the comment and suggestion. We recently published an article on the model shown in Figure 1 (Bunsick, D.A.; Matsukubo, J.; Szewczuk, M.R. Cannabinoids Transmogrify Cancer Metabolic Phenotype via Epigenetic Reprogramming and a Novel CBD Biased G Protein‐Coupled Receptor Signaling Platform. Cancers 2023, 15, 1030. https://doi.org/10.3390/cancers15041030). Also, CBD can induce (a) G-protein dependent signaling, (b) G-protein independent β-arrestin signaling, or (c) heterodimerize with AT1R/AT2R [56,57] which can crosstalk to induce signaling with MMP9 and Neu1.
- Rivas-Santisteban, R.; Lillo, J.; Raïch, I.; Muñoz, A.; Lillo, A.; Rodríguez-Pérez, A.I.; Labandeira-García, J.L.; Navarro, G.; Franco, R. The cannabinoid CB1 receptor interacts with the angiotensin AT2 receptor. Overexpression of AT2-CB1 receptor heteromers in the striatum of 6-hydroxydopamine hemilesioned rats. Experimental Neurology 2023, 362, 114319, doi:https://doi.org/10.1016/j.expneurol.2023.114319.
- Mińczuk, K.; Baranowska-Kuczko, M.; Krzyżewska, A.; Schlicker, E.; Malinowska, B. Cross-Talk between the (Endo)Cannabinoid and Renin-Angiotensin Systems: Basic Evidence and Potential Therapeutic Significance. International Journal of Molecular Sciences 2022, 23, 6350.
We have also published on the identical TLR signaling model using Bradykinin (BR2) and angiotensin II receptor type I (AT2R) which are tethered within a multimeric TOLL-like receptor (TLR) transactivation‐signaling axis, mediated by Neu1 sialidase and the glycosylation modification of TLRs [53].
- Abdulkhalek, S.; Guo, M.; Amith, S.R.; Jayanth, P.; Szewczuk, M.R. G-protein coupled receptor agonists mediate Neu1 sialidase and matrix metalloproteinase-9 crosstalk to induce transactivation of TOLL-like receptors and cellular signaling. Cellular Signalling 2012, 24, 2035-2042, doi:https://doi.org/10.1016/j.cellsig.2012.06.016.
We have research data (unpublished, manuscript in preparation) to provide evidence that CBD agonist can transactivate glycosylated receptors, including TLR and EGFR.
Specific feedback:
The introduction is somewhat arbitrary and superficial. There is a lot of information about the biased signaling of other receptors, but literature about CB1 is largely missing. It would have been more interesting to read a comprehensive and complete review on CB1 biased signaling. However, this topic is already covered in other reviews, such as Molecules. 2021 Sep 6;26(17):5413; Int J Mol Sci. 2019 Apr 13;20(8):1837.
Author response: Thank you for the comment and suggestion. We have included the following in the introduction: “Synthetic cannabinoid receptor agonists (SCRAs) are designed with the therapeutic intent of cannabinoid receptor activation [23]. However, given that CB1 receptors are GPCRs, it is highly likely they participate in bias agonism, which may lead to negative effects from further downstream signaling. A study by Patel et al. [23] found that SCRAs trigger the internalization of the CB1 receptor. In HEK cells, CB1 agonists JWH-018 and CP47,497-68 dose-dependently internalized the CB1 receptor. Another study examining various cannabinoid scaffolds found that each type exhibited biased agonism on a different signaling pathway. For example, classical cannabinoids activate the G-protein pathway, non-classical cannabinoids activate both the β-arrestin and G-protein pathway equally. However, aminoalkylindoles only cause bias agonism in the β-arrestin pathway [24]. Different classes of cannabinoids causing different bias agonism suggests the conformation of these ligands interacts differently with the receptor, inducing unique signaling effects. An additional example of cannabinoids having unique interactions is across different cell lines. Studies by Laprairie et al. [25,26] examined two cannabinoids, 2-arachidonoylglycerol (2-AG), and N-arachidonoylethanolamine (ananadamide, AEA), and found functional selectivity in in-vitro models expressing the wild-type and huntingtin protein in medium spiny projection neurons. They found that 2-AG induced biased agonism towards the Gαi/o rather than β-arrestin1 and Gαq. However, another study found that 2-AG induces biased agonism towards Gαq in rat hippocampal neurons rather than the Gαi/o found by Laprairie [27]. The results from these studies indicate bias agonism occurs at the CB1 receptor; however, the variation in ligand and cell line suggests that more research is needed to elucidate the specific mechanisms of bias agonism.”
The section 1.1 on allosteric modulation is absolutely insufficient and does not make sense as there are dozens of papers on CB1 receptor allosteric modulation, which are not put into context.
Author response: Thank you for the comment and suggestion. We have added to section 1.1 the text and citations. on CB1 receptor allosteric modulation (see above).
The logic outlined in 3.2 as to why NMBR forms dimers or interaction points with CB1 is far too speculative. It is simply not true that the observation that cannabinoids influence the signaling of glycosylated receptors means it is a transactivation via CB1. In research, often far too high concentrations of cannabinoids are used and they act rather unspecifically on many signaling pathways in vitro. Strong evidence of transactivation rather comes from CB2 receptors, where the link to TLRs is much more interesting and puzzling.
Author response: Thank you for the comment and suggestion. We have also published on the identical TLR signaling model using Bradykinin (BR2) and angiotensin II receptor type I (AT2R) which are tethered within a multimeric TOLL-like receptor (TLR) transactivation‐signaling axis, mediated by Neu1 sialidase and the glycosylation modification of TLRs [53].
- Abdulkhalek, S.; Guo, M.; Amith, S.R.; Jayanth, P.; Szewczuk, M.R. G-protein coupled receptor agonists mediate Neu1 sialidase and matrix metalloproteinase-9 crosstalk to induce transactivation of TOLL-like receptors and cellular signaling. Cellular Signalling 2012, 24, 2035-2042, doi:https://doi.org/10.1016/j.cellsig.2012.06.016.
Also, we have published on the insulin receptor regulated by the same signaling paradigm:
- Alghamdi, F.; Guo, M.; Abdulkhalek, S.; Crawford, N.; Amith, S.R.; Szewczuk, M.R. A novel insulin receptor-signaling platform and its link to insulin resistance and type 2 diabetes. Cellular Signalling 2014, 26, 1355-1368.
- Jayanth, P.; Amith, S.R.; Gee, K.; Szewczuk, M.R. Neu1 sialidase and matrix metalloproteinase-9 cross-talk is essential for neurotrophin activation of Trk receptors and cellular signaling. Cellular signalling 2010, 22, 1193-1205.
- Abdulkhalek, S.; Guo, M.; Amith, S.R.; Jayanth, P.; Szewczuk, M.R. G-protein coupled receptor agonists mediate Neu1 sialidase and matrix metalloproteinase-9 cross-talk to induce transactivation of TOLL-like receptors and cellular signaling. Cellular Signalling 2012, 24, 2035-2042, doi:https://doi.org/10.1016/j.cellsig.2012.06.016.
Haxho, Haq, Szewczuk. Cellular Signalling 43 (2018) 71–84.
Haxho et al. 2014 ijdcr-1-005
Harless et al Cells 2023 12 1332
Abdulkhalek et al. 2013 Research and Reports in Biochemistry 3-30
The conclusions do not really make sense. Why do the authors talk about functional selectivity? The concept as to why ligands would bias CB1 signaling is not well established in this review. Most data related to functional agonism and allosteric modulation of CB1 receptors are not even cited.
Author response: Thank you for the comment and suggestion. We have added the following text to the conclusion: “4. Conclusion. The concept of functional selectivity provides an exciting new direction to maximize the therapeutic potential of GPCR-targeting drugs while minimizing adverse effects. However, our understanding of how cannabinoid ligands produce biased effects is still in its infancy. A review by Morales et al. [85] highlights the structural features of GPCRs and how various intracellular conformations result in specific coupling to effector proteins, impacting the signaling pathway being activated. For example, activation of the of the transmembrane helix 6 enables Gα protein insertion into the GPCR, whereas transmembrane helix 7 favours β-arrestin coupling. However, a third conformation is possible as described by Haxho et al. [83] where the ligand heterodimerizes with NMBR before activating the Gα protein pathway. Given that CB1 is a GPCR, it is likely that functional selectivity occurs where CB1 agonists may bind to the receptor, resulting in activation of any three of these GPCR signaling pathways (see Figure 1). For this concept of biased agonism to contribute to expanding cannabinoid-based pharmaceuticals in healthcare, we must ascertain a more comprehensive understanding of the specific mechanisms of CB1 receptor functional selectivity. As presented here, investigations into the Neu1-NMBR-MMP9 signalling axis may provide unprecedented insights into these mechanisms.”
- Haxho, F.; Haq, S.; Szewczuk, M.R. Biased G protein-coupled receptor agonism mediates Neu1 sialidase and matrix metalloproteinase-9 crosstalk to induce transactivation of insulin receptor signaling. Cellular signalling 2018, 43, 71-84.
- Rozenfeld, R.; Gupta, A.; Gagnidze, K.; Lim, M.P.; Gomes, I.; Lee‐Ramos, D.; Nieto, N.; Devi, L.A. AT1R–CB1R heteromerization reveals a new mechanism for the pathogenic properties of angiotensin II. The EMBO journal 2011, 30, 2350-2363.
- Morales, P.; Goya, P.; Jagerovic, N. Emerging strategies targeting CB2 cannabinoid receptor: Biased agonism and allosterism. Biochemical pharmacology 2018, 157, 8-17.
Overall, it would be better to dedicate time to perform experiments on this hypothesis and not publish this idea too prematurely. At present, it is just an educated guess. As the reviewers properly conclude: "investigations into the Neu1-NMBR-MMP9 signalling axis may provide unprecedented insights into these mechanism". So, they should do this type of research and not just publish ideas.
Author response: Thank you for the comment and suggestion. We have research data (unpublished, manuscript in preparation) to provide evidence that CBD agonist can transactivate glycosylated receptors, including TLR and EGFR.
Also, we have research data (unpublished, manuscript in preparation) to provide evidence that CBD agonists can transactivate glycosylated receptors, including TLR and EGFR.
Comments on the Quality of English Language
The title does not really make sense. The noun "selectivity" requires the proper conjugation.
Author response: Thank you for the comment and suggestion. We have modified the title: “Functional Selectivity of Cannabinoid CB1 GPCR Transactivates Glycosylated Receptors in Health and Disease”
Reviewer 3 Report
Comments and Suggestions for Authors
While the topic of this mini-review is interesting, basic research supporting the transactivation of glycoslyated receptors by CB1 receptors is very limited. This review is therefore certainly premature as more basic research needs to be performed. There is a danger that it rather misleads the research community. The interaction of CB1 receptors with NMBR is not sufficiently substantiated by actual data. The model shown in Figure 1 is speculative and would serve as the starting point to perform more research. It is problematic when early ideas are published as scientific concepts.
Specific feedback:
The introduction is somewhat arbitrary and superficial. There is a lot of information about the biased signaling of other receptors, but literature about CB1 is largely missing. It would have been more interesting to read a comprehensive and complete review on CB1 biased signaling. However, this topic is already covered in other reviews, such as Molecules. 2021 Sep 6;26(17):5413; Int J Mol Sci. 2019 Apr 13;20(8):1837.
The section 1.1 on allosteric modulation is absolutely insufficient and does not make sense as there are dozens of papers on CB1 receptor allosteric modulation, which are not put into context.
The logic outlined in 3.2 as to why NMBR forms dimers or interaction points with CB1 is far too speculative. It is simply not true that the observation that cannabinoids influence the signaling of glycosylated receptors means it is a transactivation via CB1. In research, often far too high concentrations of cannabinoids are used and they act rather unspecifically on many signaling pathways in vitro. Strong evidence of transactivation rather comes from CB2 receptors, where the link to TLRs is much more interesting and puzzling.
The conclusions do not really make sense. Why do the authors talk about functional selectivity? The concept as to why ligands would bias CB1 signaling is not well established in this review. Most data related to functional agonism and allosteric moduation of CB1 receptors are not even cited.
Overall, it would be better to dedicate time to perform experiments on this hypothesis and not publish this idea too prematurely. At present, it is just an educated guess. As the reviewers properly conclude: "investigations into the Neu1-NMBR-MMP9 signalling axis may provide unprecedented insights into these mechanism". So, they should do this type of research and not just publish ideas.
The title does not really make sense. The noun "selectivity" requires the proper conjugation.
Author Response
REVIEWER #3
Comments and Suggestions for Authors
While the topic of this mini-review is interesting, basic research supporting the transactivation of glycoslyated receptors by CB1 receptors is very limited. This review is therefore certainly premature as more basic research needs to be performed. There is a danger that it rather misleads the research community. The interaction of CB1 receptors with NMBR is not sufficiently substantiated by actual data. The model shown in Figure 1 is speculative and would serve as the starting point to perform more research. It is problematic when early ideas are published as scientific concepts.
Author response: Thank you for the comment and suggestion. We recently published an article on the model shown in Figure 1 (Bunsick, D.A.; Matsukubo, J.; Szewczuk, M.R. Cannabinoids Transmogrify Cancer Metabolic Phenotype via Epigenetic Reprogramming and a Novel CBD Biased G Protein‐Coupled Receptor Signaling Platform. Cancers 2023, 15, 1030. https://doi.org/10.3390/cancers15041030). Also, CBD can induce (a) G-protein dependent signaling, (b) G-protein independent β-arrestin signaling, or (c) heterodimerize with AT1R/AT2R [56,57] which can crosstalk to induce signaling with MMP9 and Neu1.
- Rivas-Santisteban, R.; Lillo, J.; Raïch, I.; Muñoz, A.; Lillo, A.; Rodríguez-Pérez, A.I.; Labandeira-García, J.L.; Navarro, G.; Franco, R. The cannabinoid CB1 receptor interacts with the angiotensin AT2 receptor. Overexpression of AT2-CB1 receptor heteromers in the striatum of 6-hydroxydopamine hemilesioned rats. Experimental Neurology 2023, 362, 114319, doi:https://doi.org/10.1016/j.expneurol.2023.114319.
- Mińczuk, K.; Baranowska-Kuczko, M.; Krzyżewska, A.; Schlicker, E.; Malinowska, B. Cross-Talk between the (Endo)Cannabinoid and Renin-Angiotensin Systems: Basic Evidence and Potential Therapeutic Significance. International Journal of Molecular Sciences 2022, 23, 6350.
We have also published on the identical TLR signaling model using Bradykinin (BR2) and angiotensin II receptor type I (AT2R) which are tethered within a multimeric TOLL-like receptor (TLR) transactivation‐signaling axis, mediated by Neu1 sialidase and the glycosylation modification of TLRs [53].
- Abdulkhalek, S.; Guo, M.; Amith, S.R.; Jayanth, P.; Szewczuk, M.R. G-protein coupled receptor agonists mediate Neu1 sialidase and matrix metalloproteinase-9 crosstalk to induce transactivation of TOLL-like receptors and cellular signaling. Cellular Signalling 2012, 24, 2035-2042, doi:https://doi.org/10.1016/j.cellsig.2012.06.016.
We have research data (unpublished, manuscript in preparation) to provide evidence that CBD agonist can transactivate glycosylated receptors, including TLR and EGFR.
Specific feedback:
The introduction is somewhat arbitrary and superficial. There is a lot of information about the biased signaling of other receptors, but literature about CB1 is largely missing. It would have been more interesting to read a comprehensive and complete review on CB1 biased signaling. However, this topic is already covered in other reviews, such as Molecules. 2021 Sep 6;26(17):5413; Int J Mol Sci. 2019 Apr 13;20(8):1837.
Author response: Thank you for the comment and suggestion. We have included the following in the introduction: “Synthetic cannabinoid receptor agonists (SCRAs) are designed with the therapeutic intent of cannabinoid receptor activation [23]. However, given that CB1 receptors are GPCRs, it is highly likely they participate in bias agonism, which may lead to negative effects from further downstream signaling. A study by Patel et al. [23] found that SCRAs trigger the internalization of the CB1 receptor. In HEK cells, CB1 agonists JWH-018 and CP47,497-68 dose-dependently internalized the CB1 receptor. Another study examining various cannabinoid scaffolds found that each type exhibited biased agonism on a different signaling pathway. For example, classical cannabinoids activate the G-protein pathway, non-classical cannabinoids activate both the β-arrestin and G-protein pathway equally. However, aminoalkylindoles only cause bias agonism in the β-arrestin pathway [24]. Different classes of cannabinoids causing different bias agonism suggests the conformation of these ligands interacts differently with the receptor, inducing unique signaling effects. An additional example of cannabinoids having unique interactions is across different cell lines. Studies by Laprairie et al. [25,26] examined two cannabinoids, 2-arachidonoylglycerol (2-AG), and N-arachidonoylethanolamine (ananadamide, AEA), and found functional selectivity in in-vitro models expressing the wild-type and huntingtin protein in medium spiny projection neurons. They found that 2-AG induced biased agonism towards the Gαi/o rather than β-arrestin1 and Gαq. However, another study found that 2-AG induces biased agonism towards Gαq in rat hippocampal neurons rather than the Gαi/o found by Laprairie [27]. The results from these studies indicate bias agonism occurs at the CB1 receptor; however, the variation in ligand and cell line suggests that more research is needed to elucidate the specific mechanisms of bias agonism.”
The section 1.1 on allosteric modulation is absolutely insufficient and does not make sense as there are dozens of papers on CB1 receptor allosteric modulation, which are not put into context.
Author response: Thank you for the comment and suggestion. We have added to section 1.1 the text and citations. on CB1 receptor allosteric modulation (see above).
The logic outlined in 3.2 as to why NMBR forms dimers or interaction points with CB1 is far too speculative. It is simply not true that the observation that cannabinoids influence the signaling of glycosylated receptors means it is a transactivation via CB1. In research, often far too high concentrations of cannabinoids are used and they act rather unspecifically on many signaling pathways in vitro. Strong evidence of transactivation rather comes from CB2 receptors, where the link to TLRs is much more interesting and puzzling.
Author response: Thank you for the comment and suggestion. We have also published on the identical TLR signaling model using Bradykinin (BR2) and angiotensin II receptor type I (AT2R) which are tethered within a multimeric TOLL-like receptor (TLR) transactivation‐signaling axis, mediated by Neu1 sialidase and the glycosylation modification of TLRs [53].
- Abdulkhalek, S.; Guo, M.; Amith, S.R.; Jayanth, P.; Szewczuk, M.R. G-protein coupled receptor agonists mediate Neu1 sialidase and matrix metalloproteinase-9 crosstalk to induce transactivation of TOLL-like receptors and cellular signaling. Cellular Signalling 2012, 24, 2035-2042, doi:https://doi.org/10.1016/j.cellsig.2012.06.016.
Also, we have published on the insulin receptor regulated by the same signaling paradigm:
- Alghamdi, F.; Guo, M.; Abdulkhalek, S.; Crawford, N.; Amith, S.R.; Szewczuk, M.R. A novel insulin receptor-signaling platform and its link to insulin resistance and type 2 diabetes. Cellular Signalling 2014, 26, 1355-1368.
- Jayanth, P.; Amith, S.R.; Gee, K.; Szewczuk, M.R. Neu1 sialidase and matrix metalloproteinase-9 cross-talk is essential for neurotrophin activation of Trk receptors and cellular signaling. Cellular signalling 2010, 22, 1193-1205.
- Abdulkhalek, S.; Guo, M.; Amith, S.R.; Jayanth, P.; Szewczuk, M.R. G-protein coupled receptor agonists mediate Neu1 sialidase and matrix metalloproteinase-9 cross-talk to induce transactivation of TOLL-like receptors and cellular signaling. Cellular Signalling 2012, 24, 2035-2042, doi:https://doi.org/10.1016/j.cellsig.2012.06.016.
Haxho, Haq, Szewczuk. Cellular Signalling 43 (2018) 71–84.
Haxho et al. 2014 ijdcr-1-005
Harless et al Cells 2023 12 1332
Abdulkhalek et al. 2013 Research and Reports in Biochemistry 3-30
The conclusions do not really make sense. Why do the authors talk about functional selectivity? The concept as to why ligands would bias CB1 signaling is not well established in this review. Most data related to functional agonism and allosteric modulation of CB1 receptors are not even cited.
Author response: Thank you for the comment and suggestion. We have added the following text to the conclusion: “4. Conclusion. The concept of functional selectivity provides an exciting new direction to maximize the therapeutic potential of GPCR-targeting drugs while minimizing adverse effects. However, our understanding of how cannabinoid ligands produce biased effects is still in its infancy. A review by Morales et al. [85] highlights the structural features of GPCRs and how various intracellular conformations result in specific coupling to effector proteins, impacting the signaling pathway being activated. For example, activation of the of the transmembrane helix 6 enables Gα protein insertion into the GPCR, whereas transmembrane helix 7 favours β-arrestin coupling. However, a third conformation is possible as described by Haxho et al. [83] where the ligand heterodimerizes with NMBR before activating the Gα protein pathway. Given that CB1 is a GPCR, it is likely that functional selectivity occurs where CB1 agonists may bind to the receptor, resulting in activation of any three of these GPCR signaling pathways (see Figure 1). For this concept of biased agonism to contribute to expanding cannabinoid-based pharmaceuticals in healthcare, we must ascertain a more comprehensive understanding of the specific mechanisms of CB1 receptor functional selectivity. As presented here, investigations into the Neu1-NMBR-MMP9 signalling axis may provide unprecedented insights into these mechanisms.”
- Haxho, F.; Haq, S.; Szewczuk, M.R. Biased G protein-coupled receptor agonism mediates Neu1 sialidase and matrix metalloproteinase-9 crosstalk to induce transactivation of insulin receptor signaling. Cellular signalling 2018, 43, 71-84.
- Rozenfeld, R.; Gupta, A.; Gagnidze, K.; Lim, M.P.; Gomes, I.; Lee‐Ramos, D.; Nieto, N.; Devi, L.A. AT1R–CB1R heteromerization reveals a new mechanism for the pathogenic properties of angiotensin II. The EMBO journal 2011, 30, 2350-2363.
- Morales, P.; Goya, P.; Jagerovic, N. Emerging strategies targeting CB2 cannabinoid receptor: Biased agonism and allosterism. Biochemical pharmacology 2018, 157, 8-17.
Overall, it would be better to dedicate time to perform experiments on this hypothesis and not publish this idea too prematurely. At present, it is just an educated guess. As the reviewers properly conclude: "investigations into the Neu1-NMBR-MMP9 signalling axis may provide unprecedented insights into these mechanism". So, they should do this type of research and not just publish ideas.
Author response: Thank you for the comment and suggestion. We have research data (unpublished, manuscript in preparation) to provide evidence that CBD agonist can transactivate glycosylated receptors, including TLR and EGFR.
Also, we have research data (unpublished, manuscript in preparation) to provide evidence that CBD agonists can transactivate glycosylated receptors, including TLR and EGFR.
Comments on the Quality of English Language
The title does not really make sense. The noun "selectivity" requires the proper conjugation.
Author response: Thank you for the comment and suggestion. We have modified the title: “Functional Selectivity of Cannabinoid CB1 GPCR Transactivates Glycosylated Receptors in Health and Disease”
Round 2
Reviewer 3 Report
Comments and Suggestions for Authors
I do insist on my first verdict that this review article is premature and not really based on a lot of experimental data and thus remains speculative and in some parts misleading. For instance, in the revision the sentence "It is noteworthy
that the results of this study found that CB1R expression regulates another GPCR as it increases the relevancy of CB1R upregulation in chronic disease states and suggests that the CB1R may have interactions with other GPCR that remain to be elucidated." was added. First, it is totally unclear which study is meant, secondly, the sentence does not add anything at all as there are plenty of studies on CB1 receptors heterodimers.
It is not a review about biased signaling, where there are already many, but on the transactivation of glycosylated receptors in health and disease by CB1 receptors. However, there are almost no experimental data that would justify a review article.
The abstract is not to the point. It is strange that the first half is about GPCRs in general, then the CB1 ligands are mentioned, then the focus is on CB1 receptors. This should be improved. The sentence "Orthosteric ligands, such as THC, and allosteric ligands, including as CBD, have the potential to induce biased signaling at GPCRs." is wrong - delete the "as" CBD.
There are several trivialities that should be avoided: "Given that CB1 is a GPCR,
it is likely that functional selectivity occurs where CB1 agonists may bind to the receptor, resulting in activation of any three of these GPCR signaling pathways (see Figure 1)." It is clear that the three states described relate to the ligand binding.
Several sentences do not make sense and this reflects that the review was not constructed carefully:
"Moreover, it indicates allosteric modulation of the CB1 receptor enables activation" in the revised part.
The activation of CB1 receptors is not "a therapeutic intent"
In the legend of figure 1 it says "unctional selectivity of GPCR including CBD can induce (a) G‐protein dependent signaling (purple). This is simply not true. CBD cannot induce and has never been shown to induce signaling as it is a CB1 NAM, as shown convincingly by Pertwee and colleagues.
Comments on the Quality of English LanguageSee comments above